# Microfluidic flow-cell with passive flow control for microscopy applications

**Nicholas A. W. Bell** *, **Justin E. Molloy**

Francis Crick Institute, London, United Kingdom

* nicholas.bell@crick.ac.uk

## Abstract

We present a fast, inexpensive and robust technique for constructing thin, optically transparent flow-cells with pump-free flow control. Using layers of glass, patterned adhesive tape and polydimethylsiloxane (PDMS) connections, we demonstrate the fabrication of planar devices with chamber height as low as 25 μm and with millimetre-scale (x,y) dimensions for wide-field microscope observation. The method relies on simple benchtop equipment and does not require microfabrication facilities, glass drilling or other workshop infrastructure. We also describe a gravity perfusion system that exploits the strong capillary action in the flow chamber as a passive limit-valve. Our approach allows simple sequential sample exchange with controlled flow rates, sub-5 μL sample chamber size and zero dead volume. We demonstrate the system in a single-molecule force spectroscopy experiment using magnetic tweezers.

## Introduction

The development of simple and high-throughput methods for fluidic handling is important for many applications in microscopy and biotechnology. In particular, *in vitro* assays such as single molecule studies require techniques to exchange buffered solutions under controlled flow rates. A wide range of approaches have been developed for fabricating fluidic devices which can perform this goal varying from high-precision, lithographically designed PDMS microfluidics or glass micro-machined devices to flow-cells formed simply by gluing together coverslips using parafilm [1–10]. A thin flow-cell geometry is additionally needed for applications in which high numerical aperture (NA) optics are combined with experimental formats that require several components to be placed in close proximity to the sample. For instance, high-resolution bright-field and 4Pi fluorescence microscopies [11, 12] and force spectroscopy studies using magnetic tweezers, where the pole-pieces must come close to the specimen [13, 14].

Here, we demonstrate a low-cost system for pump-free flow control in thin fluidic devices fabricated using standard microscope coverslips and adhesive tape in combination with a PDMS exit manifold for interfacing to tubing. The method requires only simple benchtop apparatus without the need for glass drilling, complex machining or lithography. We show how the strong capillary forces in such thin glass devices can be used to create a limit valve that enables sequential sample exchange. We describe the construction of single and multi-stream

**Data Availability Statement:** All relevant data are within the paper and its Supporting Information files.

**Funding:** NAWB was supported by an AstraZeneca-Crick collaborative grant (FC001632) and also by the Francis Crick Institute, which

receives core funding from CRUK (FC001119), MRC (FC001119) and the Wellcome Trust (FC001119). The funders did not have any additional role in the study design, data collection and analysis, decision to publish, or preparation of the manuscript. AstraZeneca provided support in the form of funding for the research and salary for an author [NAWB]. AstraZeneca reviewed the final draft of the paper and approved its submission without amendments. The specific roles of NAWB are articulated in the 'author contributions' section.

**Competing interests:** AstraZeneca contributed to the funding of this project and the salary of NAWB. This does not alter our adherence to PLOS ONE policies on sharing data and materials. The authors declare no other competing interests.

devices and show repeatable and stable solution flow in a single-molecule force spectroscopy assay.

## Results

### Device design principles

The device is assembled in a sequence of steps as illustrated in Fig 1. A detailed list of the exact materials we used for the various components is given in the Materials and methods section at the end of the paper. First, channels are cut into double-sided adhesive tape. Different tapes can be used depending on the required channel thickness. The channels in the tape can be made either by hand using a scalpel blade or preferably using an inexpensive programmable cutting machine (e.g. Cricut Explore) or a laser cutter which allow complex designs to be made in a reproducible way. For illustrative purposes we show a simple, single-channel device. After making the channel, the tape is affixed to the shorter, upper coverslip (Fig 1A). Any overhanging edges of tape are trimmed away using a scalpel blade and the coverslip is aligned and pressed against the longer, lower coverslip (Fig 1B). At this point the two coverslips should be placed on a flat surface so that even pressure can be applied to ensure a good seal is made across the full width of the tape leaving the empty channel that runs across the full length of the upper coverslip.

Next, a tubing connection is created at the outlet end using a combination of double-sided adhesive tape and PDMS. A small section of double-sided tape (~5 mm x 6 mm) is cut and a central, circular hole is made using a 1.5 mm diameter biopsy punch. The tape used here must have a silicone-based adhesive in order to make a good seal with PDMS. The tape is carefully positioned so that the central hole just overlaps with the exit port of the flow-cell channel (Fig 1C). This allows smooth flow between the channel and tubing in the finished device and therefore reduces the chances of air bubble formation [15]. A magnifying glass stand or dissection microscope facilitates alignment with the exit port. The air gap formed at the junction between the tape and the top coverslip is filled by adding a small amount of uncured PDMS mixture at either end of the air gap and allowing the PDMS to penetrate and seal the air gap (Fig 1D). This stage requires careful timing to ensure that the PDMS doesn't run into the channel. To reduce the flow speed of the PDMS it helps to partially polymerize the PDMS by heating it for 4 minutes at 80˚C which increases its viscosity. Once the PDMS has reached the edge of the hole, it is finally cured by placing the device on a hot plate for two minutes at 100˚C.

A section (~5 mm x 6 mm) of cured PDMS is cut with a scalpel from a 4 mm thick sheet that had been previously poured and left to cure in a plastic petri dish. Then a central, circular hole is punched through using a 1.5 mm diameter biopsy punch. The PDMS block is then positioned so that the central hole aligns with the corresponding hole in double-sided tape stuck across the exit port (Fig 1E). The PDMS block is fixed in place by applying even pressure with the flow cell lying on a hard flat surface. The same steps (Fig 1C–1E) can be performed at the inlet of the channel to form another connection for tubing. Alternatively, as shown in Fig 1F, PDMS can be manually painted and cured around the channel end to form a hydrophobic barrier for fluid. This can easily be achieved without getting PDMS in the inlet since a small amount of PDMS applied to the top coverslip will flow to the coverslip edge boundary but not flow over the edge due to surface tension. This method enables a reduction in dead-volume and, as will be shown in Figs 2 and 4, creates a convenient method for passive control of flow when combined with gravity perfusion. The manual painting method (Fig 1F) enables an inlet reservoir size of ~30 μL. For experiments requiring larger sample reservoirs, eg when long term, high flow rates are needed, it is straightforward to use PDMS to affix a small reservoir, for instance, one made from the large end of a plastic pipette tip. At the outlet, tubing of the

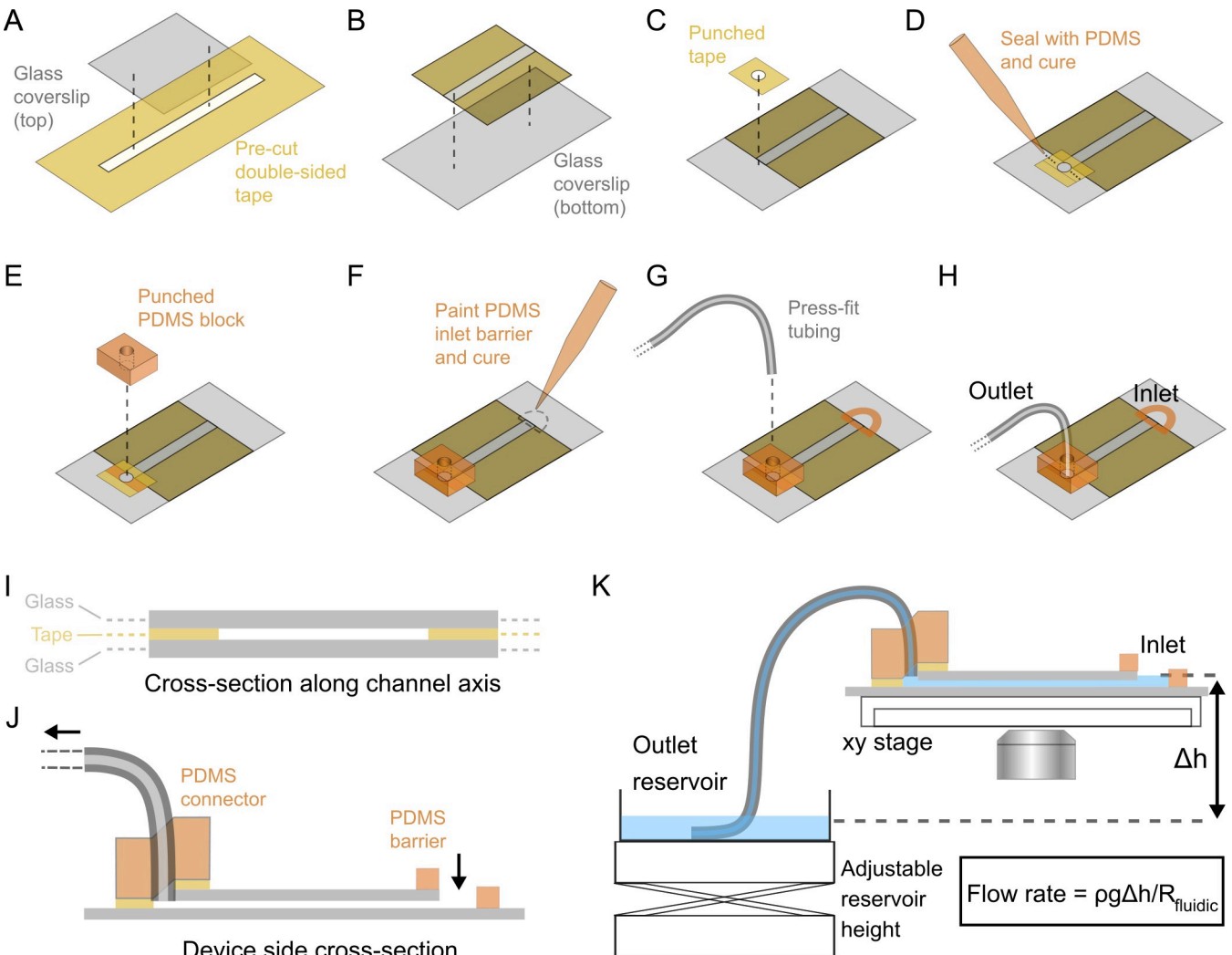

**Fig 1. Fabrication workflow and experimental setup.** (A-H) Steps in fabricating the flow-cell. The steps are carried out left to right in two rows. (I) Cross-section along channel axis. The schematic is to scale for 1.5 mm channel width, 81 μm tape thickness and no. 1 thickness coverslips. (J) Cross-section through side of finished device showing the bottom and top glass coverslips, the PDMS barrier forming the inlet and the PDMS connector used for the outlet tubing. (K) Experimental setup for gravity driven perfusion. The device is secured on an (x,y) stage using stage clips and the outlet tubing leads to a large reservoir positioned on a height adjustable platform. The flow rate is adjusted by changing the height between the inlet and outlet reservoir or by changing the fluidic resistance of the system by adjusting the flow cell and tubing design geometry.

appropriate diameter can then be inserted by press-fit into the hole in the PDMS block to form a tight seal (Fig 1G).

Fig 1H shows the finished device layout and Fig 1I and 1J show cross-sections of the device. In Fig 1K, we show the device on a typical microscope rig. The bottom coverslip is secured by stage clips and the outlet tubing leads to a wide fluid reservoir eg a petri dish. The flow rate, Q, is given by

$$Q = \frac{\rho g \Delta h}{R_{fluidic}}, \qquad (1)$$

where $\rho$ is the density of the fluid, g is the acceleration due to gravity, $\Delta h$ is the height difference between inlet and outlet levels and $R_{fluidic}$ is the total fluidic resistance of the device. To

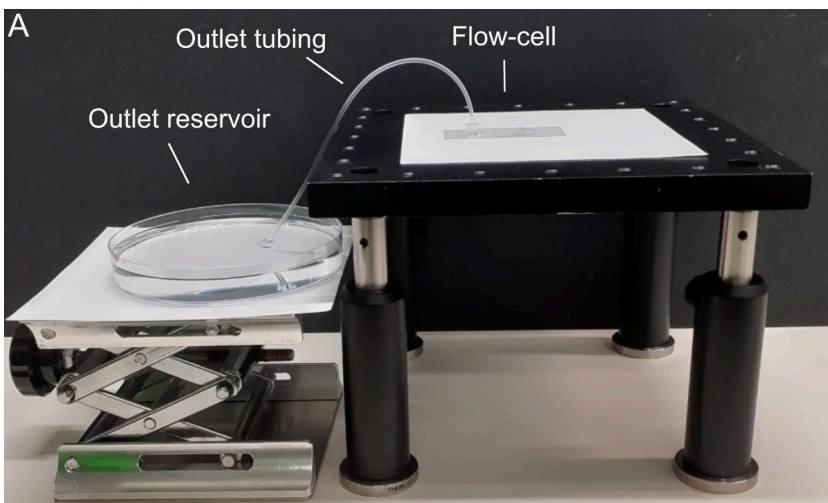

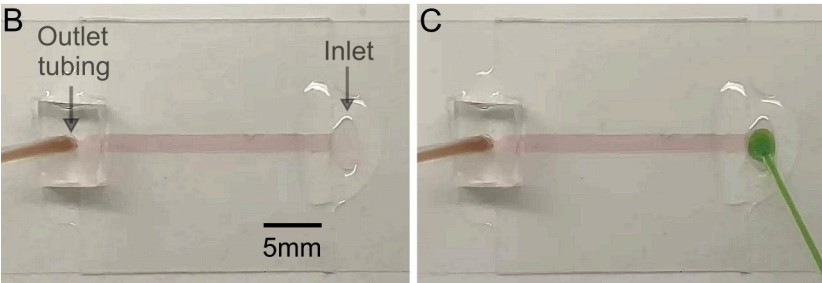

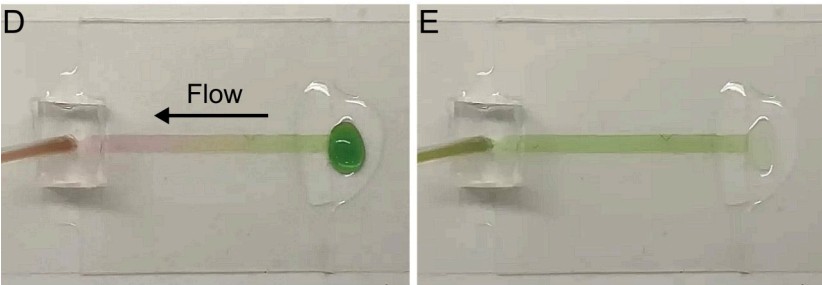

**Fig 2. Simple sequential exchange of solutions with a passive stop due to capillary action.** (A) Apparatus for generating gravity-driven flow. (B) The device was filled with purple dye solution to show the channel. The outlet tubing connects to an outlet reservoir 4 cm below the level of the inlet. (C) 10 µL green dye solution is added at the inlet using a pipette. (D) Solution flows through the device until (E) when the solution reaches the entrance and passively stops without experimenter input.

calculate the fluidic resistance, it is important to determine whether the flow is laminar. The characteristics of a fluid flow in a channel i.e. laminar vs turbulent flow can be determined by calculation of the Reynold's number (Re)–a dimensionless quantity which gives the ratio of inertial to viscous forces:

$$\mathrm{Re} = \frac{\mathrm{v}l\rho}{\eta}, \tag{2}$$

where v is the fluid velocity, l is the characteristic length scale of the system, ρ is the solution density and η is the dynamic viscosity. In our experiments, even at the highest flow velocities used (see Fig 4), the Reynolds number is significantly less than 2000 –the approximate transition point between laminar and turbulent flow [16]. Eg at v = 30 mm.s$^{-1}$, l = 0.1 mm, ρ = 1000 kg.m$^{-3}$, η = 0.001 Pa.s the value of Re = 3. The total fluidic resistance of the device is then determined by the dimensions of the channel and tubing and is given by [17]

$$R_{fluidic} = R_{tubing} + R_{channel} = \frac{128\eta L}{\pi d^4} + \frac{12\eta a}{b^3 w}, \tag{3}$$

where L is the length of tubing, d is the tubing diameter, a is the channel length, b is the channel height and w is the channel width.

By choosing the dimensions of the tubing and channel, the desired flow rate can be achieved. We also use an adjustable platform so that the height difference between inlet and outlet reservoir levels, Δh, gives a control of flow rate, over a useful range, during an experiment. We typically use 81 μm thick acrylic tape for the flow channel together with no.1 coverslips resulting in a total device thickness of ~380 μm. The minimum thickness we have achieved with adhesive transfer tape of 25 μm thickness combined with class 0 coverslips is just ~250 μm.

## Flow control using capillary forces

The use of glass for the top and bottom layers of the device, together with the sub-200 μm channel height, creates strong capillary forces acting in the channel. If this capillary action is greater than the gravity-feed driving force then, once the inlet reservoir is exhausted, solution flow will cease but the channel will remain full of solution. A simple model predicts the maximum height between inlet and outlet reservoir levels, Δh, where capillary action will be able to prevent flow continuing beyond the entrance: The Laplace pressure, ΔP, at the interface between the fluid in the capillary and the air, for a parallel plate geometry, is given by

$$\Delta P = \frac{2\gamma \cos\theta}{b}, \tag{4}$$

where γ is the surface tension, θ is the contact angle and b is the height between the two parallel plates (in our case the tape thickness). Capillary action will hold the solution if $\frac{2\gamma\cos\theta}{b} \geq \rho g\Delta h$. Taking values for water in a typical flow chamber where: b = 81 μm, γ = *0.07 N.m$^{-1}$*, θ = 0° (i.e. complete wetting), ρ = 1000 kg.m$^{-3}$, g = 10 m.s$^{-2}$ predicts that the system will passively stop as long as Δh ≤ 17.5cm. In practice, we measured the limiting height to be Δh ~ 11cm for deionised water when the flow chamber was setup with tubing as in Figs 2 and 4C. This sets a maximum for the flow-rate achievable, for a given device geometry, when a passive flow-stop system is desired. We note that the maximum height for a passive flow-stop scales as the inverse of the channel height whereas the flow rate scales as the cube of the channel height according to Eq (3). Therefore, experiments requiring a high flow rate will benefit from increasing the channel height. This however must be balanced by other factors which favour a smaller channel height such as the need to have a low chamber volume and the time dependence in flow rate caused by having a small height between the inlet and outlet levels (see later section).

In Fig 2, we show the essential components of the apparatus and demonstrate the passive stop mechanism. Fig 2A shows a side view of the apparatus used. The device sits on a platform above the large outlet reservoir. Tubing with internal diameter of 0.51 mm is connected to the PDMS manifold of the device and acts as the connection between the device and outlet

reservoir. Fig 2B–2D shows a bird's eye view of the flow channel. The channel width and length were cut precisely using a Cricut Explore Air 2 cutting machine giving overall dimensions w = 1.5 mm, a = 22 mm, b = 81 μm (total volume = 2.7 μL). The device was filled with purple dye solution (Fig 2B) and the outlet reservoir level set to a height Δh = 4cm below the inlet. After addition of 10 μL green dye solution (Fig 2C), the solution flows through continuously (Fig 2D) until it stops at the entrance (Fig 2E). This sequential sample addition can be repeated multiple times without air bubble formation (S1 Video) and is therefore a convenient alternative to more complex fluidic valve systems which are typically used for switching between solutions. One potential drawback of this system is that the open entrance at the inlet means that evaporation will result in a small residual flow as solution is drawn up towards the entrance by capillary action to compensate for the evaporative loss. Using a device with an outlet sealed with epoxy resin and on open inlet, we measured the loss due to evaporation to be ~0.5 μL.hr$^{-1}$ ie ~ 0.1 nLs$^{-1}$ in a room at 22°C and 45% relative humidity. This small rate of back-flow is negligible for most applications and could be further reduced by decreasing the channel width at the inlet.

## Multistream device fabrication

Multiple streams are useful in certain applications where solutions must be separated but simultaneously accessible on the same device. For instance, in single-molecule experiments a technique for assembling DNA with a dual-beam optical trap relies on creating separate streams for beads and DNA [18]. Our method is readily extended to the fabrication of devices with multiple channels by programmable cutting of the double-sided tape. Fig 3A shows an example design where four separate channels converge to a single channel. PDMS is manually

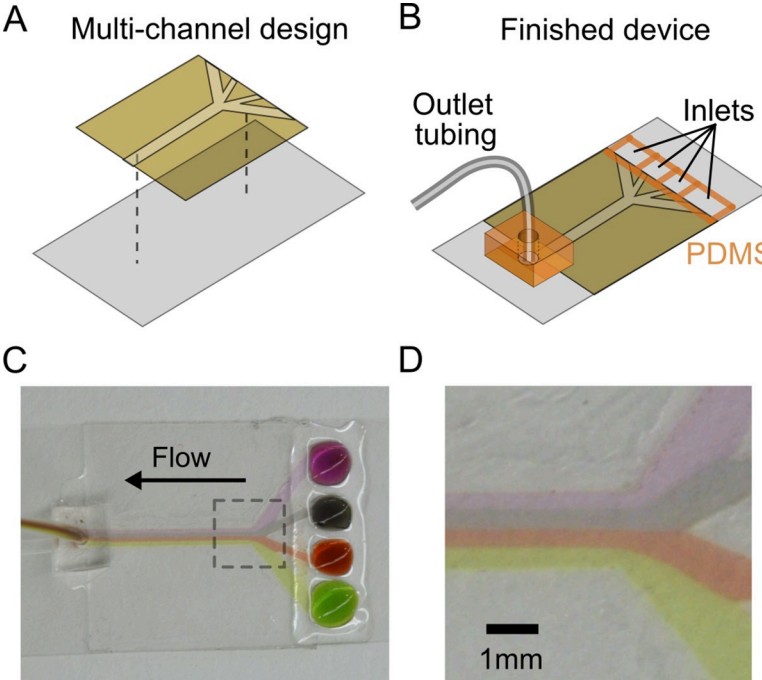

**Fig 3. Multi-channel flow device.** (A) Computer aided design is used to fabricate a pattern in double-sided tape using a programmable cutting machine. (B) The device is assembled as in Fig 1 but with PDMS barriers for four separate inlets. (C) Bird's eye view of device with multiple streams in laminar flow. (D) Magnified image defined by the dashed box in (C), showing the four separate laminar flow streams.

painted to separate the four inlets (Fig 3B). Fig 3C and 3D show images of the assembled device with four separate dyes used to image the flow streams. The low Reynolds number creates a laminar flow without cross-currents perpendicular to the channel axis so that mixing is mainly determined by diffusion between the streams.

### Example force spectroscopy application

To demonstrate the application of our method in a microscopy study, we used a single channel device for a magnetic tweezers DNA stretching experiment. Fig 4A shows a schematic of the experiment where a pair of neodymium magnets are positioned above the surface of the flow cell that is mounted on the stage of an inverted microscope. Magnetic beads are attached to the bottom coverslip via single DNA molecules (7.9 kbp long). When the magnets are lowered in close proximity to the flow cell, the magnetic field gradient causes the beads to rise from the coverslip surface until the DNA molecules are pulled taut and extended close to their contour length of 2.7 μm (Fig 4B). When buffer is added to the inlet reservoir, solution flow

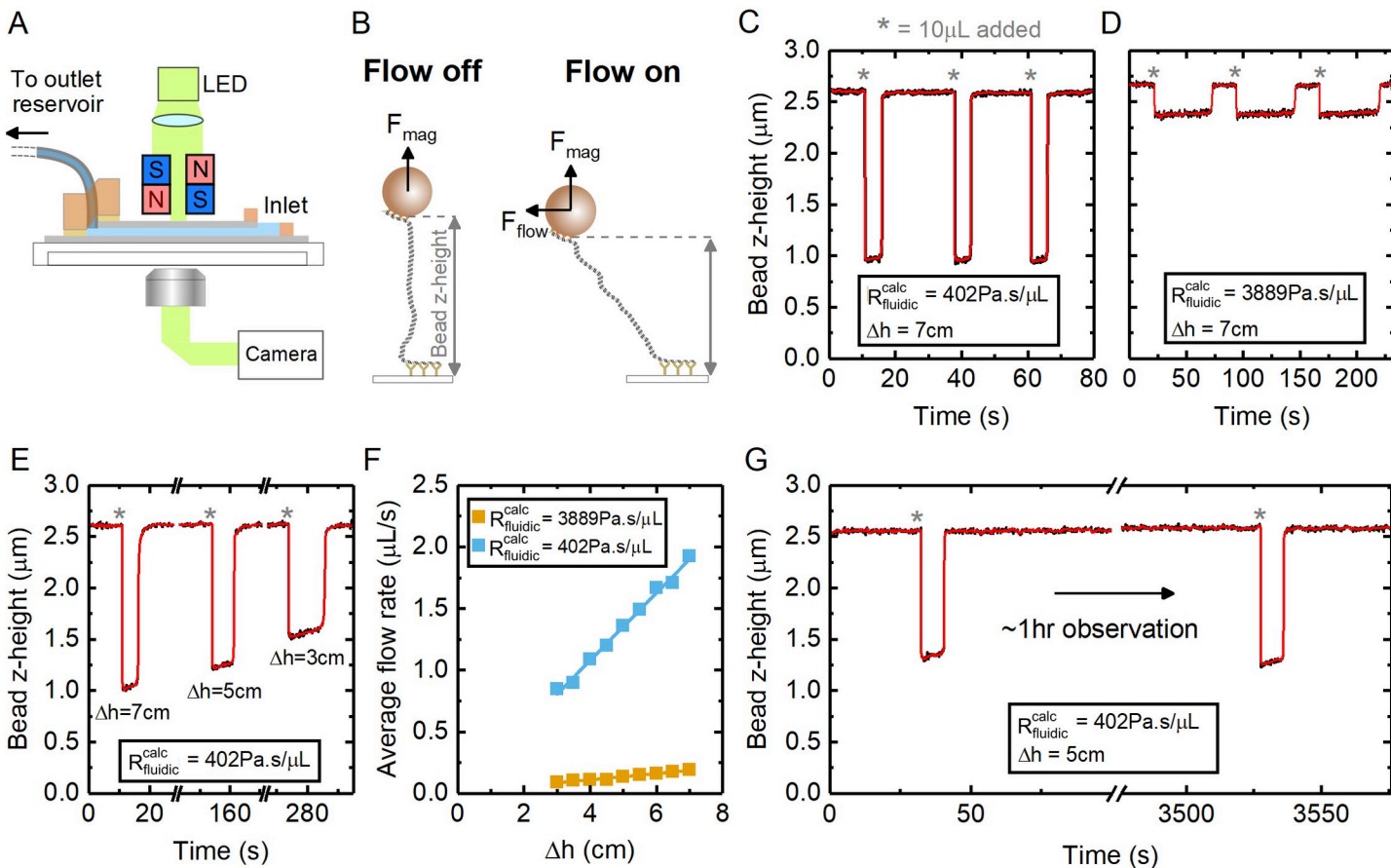

**Fig 4. Controllable addition of samples in a single-molecule DNA stretching experiment.** (A) Experimental schematic showing flow-cell together with magnets positioned at close proximity to the top cover slide to apply a fixed force. (B) DNA is tethered to the surface using a magnetic bead. At zero flow, the magnetic force acts away from the surface and stretches the DNA. When solution is added, the bead position is deflected due to the solution flow parallel to the channel axis. (C) Bead z-height as a function of time for the device using 20 cm of 0.51 mm inner diameter tubing with fluidic resistance calculated as 402 Pa.s.μL$^{-1}$. The asterisks indicate timepoints where 10 μL of solution was manually added at the inlet. (D) Same experiment as (C) after adding 25 mm of 0.127 mm diameter narrow bore tubing to increase the calculated fluidic resistance to 3889 Pa.s.μL$^{-1}$. (E) Using the device in the same configuration as (C)– 10 μL samples of solution were added and the height between the inlet and outlet reservoir levels was varied as indicated. (F) Comparison of flow rates through the device for the two different fluidic resistance configurations as a function of inlet and outlet reservoir height difference. The orange linear fit gave a value of R$_{fluidic}$ = 3990 Pa.s.μL$^{-1}$ and the blue linear fit gave a value of R$_{fluidic}$ = 366 Pa.s.μL$^{-1}$. (G) Experimental recording showing the addition of 10 μL solution samples separated by an interval of ~1 hr.

commences and Stokes' drag on the beads causes them to undergo a pendulum-like swing along the direction of flow reducing their height above the coverslip (z-displacement). The flow rate is monitored, in real-time, by measuring bead position with video image analysis.

We assembled a device with channel dimensions of w = 1.5 mm, a = 22 mm, b = 81 μm. This was connected to 20 cm of 0.51 mm inner diameter tubing giving a total fluidic resistance (from Eq (3) using η = 8.9x10$^{-4}$ Pa.s) of $R_{fluidic}$ = 402 Pa.s.μL$^{-1}$. Fig 4C shows the change in bead z-height when 10 μL buffer solution (PBS + 0.2 mg.mL$^{-1}$ BSA) is manually pipetted onto the device inlet at three timepoints. The force due to the magnetic field, $F_{mag}$, was held constant at 7 pN in all measurements. The bead is deflected by the flow when solution is added and quickly returns to its original height once the inlet is depleted. Importantly, the experiment shows that the flow is repeatable and stable with minimal residual flow. To reduce the flow rate, we added a 25 mm length of 0.127 mm inner diameter tubing, in series with the 20 cm of 0.51 mm inner diameter tubing, by using a simple connector made from a 4 mm thick block of cured PDMS with a press fit hole made using a biopsy punch. This narrow bore tubing substantially increases (by about 10-fold) the fluidic resistance according to Eq (3) to $R_{fluidic}$ = 3889 Pa.s.μL$^{-1}$. Fig 4D shows the changes in bead z-height when three samples of 10 μL buffer solution are added and, as expected, a substantial reduction in flow rate is observed.

The flow rate can also be adjusted, within a limited range, by changing the height of the outlet reservoir. Fig 4E shows an experiment where the outlet reservoir height was adjusted between sample additions. We used flexible (Tygon) tubing between the channel outlet and outlet reservoir to help minimize mechanical perturbation to the system when the outlet reservoir height is changed. In Fig 4F we quantify the observed average flow rate as a function of height between the inlet and outlet reservoirs for the device with and without the narrow bore tubing. This shows the expected linear dependence, according to Eq (1), with least-squared fits giving values of resistance within 10% of the resistance calculated from Eq (3). In general, to create a constant flow rate it is desirable to have the maximum allowable distance between the inlet and outlet reservoirs since the variation in the droplet size at the inlet during flow can create a small time varying component to the height difference between the inlet and outlet reservoir. This can be observed as a small reduction in bead deflection (and therefore flow rate) during the course of solution flow–for instance in Fig 4E. Experiments requiring more precise flow rate control may benefit from addition of a flow rate sensor with a feedback controlled piezo stage for the outlet reservoir to correct for the small, time-varying differences in reservoir heights during flow.

The device permits stable observation for long periods of time between solution additions– an important pre-requisite for many microscopy work-flows. Fig 4G shows an example experiment where two 10 μL solution samples were added with approximately a 1 hr time interval between additions. The magnetic bead position remained steady between these timepoints and the flow rate (as shown by the time the bead was deflected) is matched. This demonstrates that capillary action stably holds the solution in the chamber—without air bubble formation— enabling long term data acquisition.

## Multiplexing and flow reproducibility

To demonstrate high experimental throughput and test device reproducibility, we made a design with four channels programmably cut to fit the coverslip area. We also programmably cut the silicone adapter tape to feature four circular holes for the channels together with five square holes which enable PDMS addition for sealing each channel. The key steps in assembly of the device are shown in Fig 5A–5F. Fig 5G shows an image of a finished device filled with dye solutions.

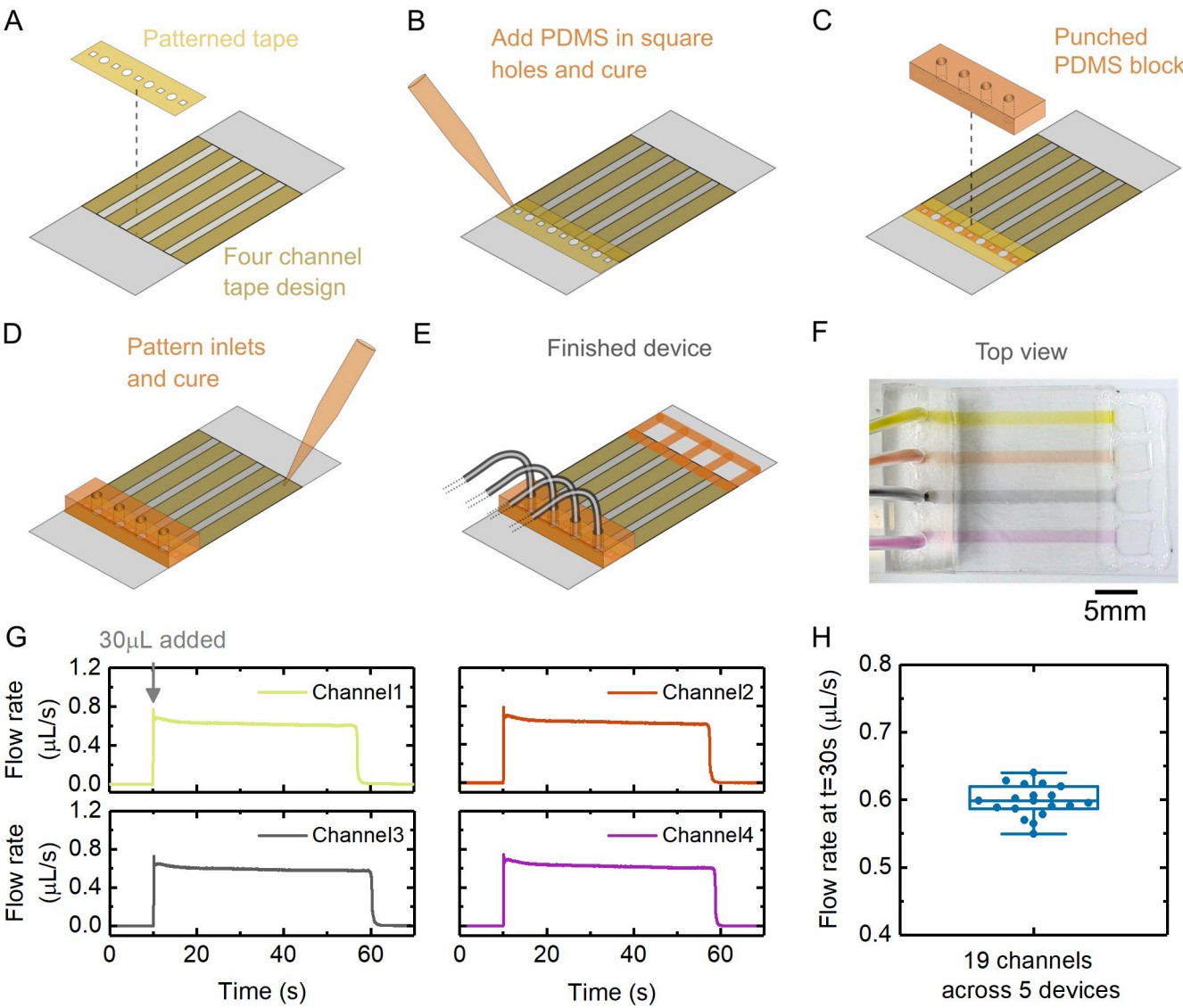

**Fig 5. Multiple channel design and reproducibility test.** (A-E) Key steps in fabricating the flow chamber. The steps are carried out left to right in two rows. (F) Image of flow chamber filled with four different dye solutions. (G) Flow sensor recordings from four channels on one device. Each channel was filled with deionised water and 30 μL deionised water was added at t = 10s. There is a brief transient at the beginning of the flow due to the fluidic capacitance of the device. (H) Flow rate values measured at t = 30s for 19 channels across five devices (one channel was blocked with PDMS and therefore not measured).

We made five devices using this design and filled each channel with deionised water. To test the flow rate reproducibility between channels, a flow rate sensor (Fluigent, FLU-M-D) was connected to the outlet tubing to record the flow at 50 Hz sampling rate when 30 μL deionised water solution was added to the inlet. The same exact tubing was used for each channel (see Materials and methods). The height between the inlet and outlet reservoirs was $\Delta h$ = 7cm. Fig 5G shows examples of the flow rate recorded by the sensor from four separate channels on one device when 30 μL deionised water was added at the inlet. In Fig 5H, we plot the recorded flow rate at t = 30s (ie 20s after solution addition) from 19 channels measured across five devices. This shows the high reproducibility of the technique from channel to channel with a median flow rate of 0.60 $\mu L.s^{-1}$ and interquartile range of 0.03 $\mu L.s^{-1}$.

## Discussion

In summary, we have presented a novel method for high-throughput fabrication of thin, glass flow chambers. The devices are low-cost and easy to manufacture, as they do not require complex fabrication techniques such as glass drilling or lithography. This means they can be single use, disposable items ideal for many research laboratory work-flows. In contrast to PDMS lithography-based microfluidics, our method enables rigid, ultrathin devices. These flow cells therefore have particular application in microscopy studies which require close access to the sample e.g. with high numerical aperture objectives and condenser optics or for techniques such as magnetic and optical tweezers. Since the contact angle for water on glass is significantly lower than water on PDMS, the use of glass for the top and bottom layers of the device also enables strong capillary stop valves. The (x,y) resolution of the channels formed using adhesive tape is, however, lower than what is possible with lithography-based microfluidics. PDMS lithography standardly enables feature sizes down to the micron level [19]. From our experience, the minimal channel width that can be fabricated, in adhesive tape, with a programmable mechanical cutting blade is ~500 μm. A recent demonstration showed that a laser cutter can be used to make channels in adhesive tape with widths down to ~100 μm [20].

We also presented a method for controlling flow rate using gravity feed and a sequential sample addition technique which exploits the capillary action of the device to automatically arrest flow when solution supply in the inlet reservoir is exhausted. Our approach for flow control using a capillary valve has a number of valuable aspects. The lack of tubing on the inlet side results in zero dead volume and the sample chamber size is easily fabricated to be <5 μL which is important for precious samples. The equipment costs and expertise are minimal since no valves, syringe pumps or pressure regulators are needed. There is also minimal mechanical perturbation when solutions are exchanged enabling continuous tracking of objects such as magnetic beads. These features make this technique for device fabrication and flow control a useful tool for microscopy studies.

## Materials and methods

### Equipment and consumables for device construction

For channels made in the tape by hand, the only equipment needed is a hot plate to cure the PDMS at 100°C and a magnifying stand or dissection microscope for aligning the layers of the device. However, it is preferable to use a programmable cutting machine to make reproducible designs particularly for the multiple channel devices. We used a Cricut Explore Air 2 machine which retails for ~$250.

The following basic consumable items are needed: (1) Top and bottom coverslips. (2) Double-sided tape for channel formation. (3) Silicone based double-sided tape for PDMS connection. (4) PDMS. (5) Syringe and tips for dispensing PDMS and priming device. (6) Biopsy punch. (7) Tubing. (8) Petri dish for outlet reservoir. We used the following specific materials: Coverslips (thickness #1 and #0) were purchased from ThermoFisher (Menzl-Glasser). 22 mm x 22 mm size coverslips were used for the top layer and 22 mm x 50 mm size coverslips were used for the bottom layer. To form the channels, we have successfully used either 81 μm thickness acrylic based tape (Adhesives Research, AR90445), 25 μm thickness silicone based tape (Adhesives Research, AR8026) or 50 μm thickness acrylic based tape (ThorLabs, OCA8146-2). The 25 μm and 50 μm tapes are so-called adhesive transfer tapes which do not have a central film sandwiched between the adhesive. These types of tape are prone to tear when machined with a mechanical cutter and a laser cutter gives more consistent results. For the 81 μm tape (used in all devices shown in the manuscript), the channels were cut to designed dimensions

using a Cricut Explore Air 2. A 140 μm thickness silicone based tape (Adhesives Research, AR90880) was used for bonding the PDMS and glass (Fig 1C–1E) in all devices shown. Alternatively we have also had success using 120 μm thickness differential adhesive double-sided tape (Grace Bio-labs SecureSeal, GBL620001, Sigma-Aldrich) with the silicone side used for PDMS and the acrylic side used for glass. Both the 140 μm and 120 μm thickness tapes could also be used for the channels. PDMS (Sylgard 184, Dowsil) was used at 10:1 ratio of elastomer:curing agent. Aliquots of mixed PDMS can be stored at -20˚C for up to a month without curing for convenient preparation ahead of time. The PDMS was manually applied in (Fig 1D and 1F) by using a plastic 5 mL syringe filled with mixed PDMS connected to a plastic pipette tip or a 23ga blunt end ¼" tips (0401-08-000094, Ellsworth Adhesives). 1.5 mm biopsy punches (BP15F, Selles Medical) were used for punching holes in PDMS and used in combination with Tygon ND-100-80 Microbore tubing ID = 0.02" = 0.51mm, OD = 0.06" = 1.5 mm (Cole-Parmer). Each finished device was initially filled with solution by attaching a 100 μL Hamilton syringe to the outlet tubing using 23ga blunt end ¼" tips (0401-08-000094, Ellsworth Adhesives) and drawing solution through manually. For the high fluidic resistance device (Fig 4C), a 25 mm section of PEEK tubing with ID = 0.005" and OD = 1/16" (13069892, Kinesis 1535, Fisher Scientific) was connected to the end of the Tygon tubing using a connector made by punching a 1.5 mm hole in a 4 mm thick section of cured PDMS and inserting the tubing ends into this connector. For biopsy punching steps, it helps to place the PDMS to be punched on top of a piece of waste PDMS material to allow the biopsy punch to penetrate fully without damaging the tip. Vector graphics files for the various designs of tape are available from the authors upon request.

## Magnetic tweezers

For magnetic tweezers measurements, the bottom coverslip was coated with 2% collodion (Sigma-Aldrich) by spin-coating. The device was assembled and then coated with anti-digoxigenin antibodies (Roche) before passivation with BlockAid (ThermoFisher). Magnetic beads were incubated with a DNA construct consisting of a central 7.9 kbp insert and two "handles" at either end made by PCR incorporation of digoxigenin and biotin nucleotides. The magnetic beads were flowed into the device and incubated for five minutes before flowing through PBS (Gibco 10010–031) + 0.2 mg.mL$^{-1}$ BSA which had been filtered with a 0.2 μm syringe filter. The bead position was tracked by video microscopy using a Nikon TE-2000 microscope, high powered LED illuminator and 60x Nikon air objective. The z-height of the bead was computed by using a look-up table for the diffraction ring pattern [21]. The microscope drift was corrected by subtracting the position of a fiducial silica bead which was bound to the surface. Each solution addition (Fig 4C–4E and 4G) consisted of 10 μL PBS + 0.2 mg.mL$^{-1}$ BSA. The bead position was acquired at 20 Hz imaging rate. In Fig 4C–4E and 4G the black trace shows the data with a 3-point median filter and the red trace shows a 10-point median filter. The average flow rate shown in Fig 4F was calculated by determining the time that the bead was deflected and dividing by the volume added.

## Flow rate sensor measurements

The flow rate was measured for each channel individually using a commercial flow rate sensor (Fluigent, FLU-M-D). Deionised water was used. The same combination of tubing in series was used in each recording– 1.6 cm of Tygon ND-100-80 (ID = 0.02", OD = 0.06") for connecting to the PDMS block, then 12 cm PEEK tubing (ID = 0.5mm, OD = 1/16") and 10.1 cm PEEK tubing (ID = 0.01", OD = 1/32") for connecting to the flow sensor. Since the exact dimensions of the flow sensor are not known, we did not estimate the expected fluidic resistance in this measurement.

## Supporting information

**S1 Video. Repeated sample exchange with passive flow stop.** S1 Video shows the raw video from which the images of Fig 2 are made. A device is filled with purple dye solution and 10 μL of dye solution are added sequentially alternating between green and purple dyes.
(MP4)

## Acknowledgments

The authors thank Jehangir Cama for helpful discussions, Daniel Burnham for help with using a Cricut Explore programmable cutting device, Gregory Mashanov for magnetic tweezers software and George Konstantinou for assistance with flow sensor recordings.

## Author Contributions

**Data curation:** Nicholas A. W. Bell.

**Formal analysis:** Nicholas A. W. Bell.

**Funding acquisition:** Justin E. Molloy.

**Investigation:** Nicholas A. W. Bell.

**Methodology:** Nicholas A. W. Bell, Justin E. Molloy.

**Project administration:** Justin E. Molloy.

**Resources:** Justin E. Molloy.

**Software:** Nicholas A. W. Bell, Justin E. Molloy.

**Supervision:** Justin E. Molloy.

**Validation:** Nicholas A. W. Bell, Justin E. Molloy.

**Visualization:** Nicholas A. W. Bell.

**Writing – original draft:** Nicholas A. W. Bell.

**Writing – review & editing:** Nicholas A. W. Bell, Justin E. Molloy.

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
