## [Decision Letter · Decision Letter 0]

1 Oct 2020

PONE-D-20-26561

Laminar flow chambers with pump-free flow control for microscopy applications

PLOS ONE

Dear Dr. Bell,

Thank you for submitting your manuscript to PLOS ONE. After careful consideration, we feel that it has merit but does not fully meet PLOS ONE’s publication criteria as it currently stands. Therefore, we invite you to submit a revised version of the manuscript that addresses the points raised during the review process.

We look forward to receiving your revised manuscript.

Kind regards,

Ji Yi

Academic Editor

PLOS ONE

Journal Requirements:

2. Thank you for stating the following in the Financial Disclosure section 

"N.A.W.B. is supported by an AstraZeneca-Crick collaborative grant (FC001632) and also by the Francis Crick Institute, which receives core funding from CRUK (FC001119), MRC (FC001119) and the Wellcome Trust (FC001119). The funders had no role in study design, data collection and analysis or preparation of the manuscript.".

 We note that one or more of the authors have an affiliation to the commercial funders of this research study : 'AstraZeneca-Crick'

Additional Editor Comments (if provided): Please address the reviewer's comment on the novelty of the work, and provide more evidence on the superiority of the manufacturing in compared with PDMS lithography based technologies. 

Reviewers' comments:

Reviewer's Responses to Questions

**Comments to the Author**

1. Is the manuscript technically sound, and do the data support the conclusions?

Reviewer #1: Yes

2. Has the statistical analysis been performed appropriately and rigorously? 

Reviewer #1: N/A

3. Have the authors made all data underlying the findings in their manuscript fully available?

Reviewer #1: Yes

4. Is the manuscript presented in an intelligible fashion and written in standard English?

Reviewer #1: Yes

5. Review Comments to the Author

Reviewer #1: In this manuscript the authors reported the design of a lab-fabrication friendly flow chamber. Such flow chamber can be prepared without the requirement of microfabrication facilities, or other workshop instruments. Overall the authors did a great job describing the detailed design concept regarding the chamber, and the following proof-of-concept application study is well planned. However, there are still several issues that authors need to address, and some additional information needed to support all their claims. Based on its current status, I would recommend “revision”.

1) The author claimed that the chamber can be made in a high-throughput fashion, but didn’t provide much support to demonstrate the chamber to chamber variability, results reproducibility, etc. and how much “faster” can it be when compared to the PDMS lithography based technologies?

2) How does the chamber perform when compared side-by-side with a similarly designed lithography designed microfluidic chamber? What’s the novelty of the chamber performance-wise?

3) Would the material difference between the chamber wall material (the tape) and the top/bottom material (glass) create any change in the flow pattern? And how would the different types of the tape material impacts the flow?

6. PLOS authors have the option to publish the peer review history of their article (what does this mean?). If published, this will include your full peer review and any attached files.

Reviewer #1: No

---

## [Author Response · Author response to Decision Letter 0]

11 Nov 2020

Please see attached word document.

---

## [Editor Report · Decision Letter 1]

3 Dec 2020

Microfluidic flow-cell with passive flow control for microscopy applications

PONE-D-20-26561R1

Dear Dr. Bell,

We’re pleased to inform you that your manuscript has been judged scientifically suitable for publication and will be formally accepted for publication once it meets all outstanding technical requirements.

Kind regards,

Ji Yi

Academic Editor

PLOS ONE
---

## [Editor Report · Acceptance letter]

7 Dec 2020

PONE-D-20-26561R1 

Microfluidic flow-cell with passive flow control for microscopy applications 

Dear Dr. Bell:

I'm pleased to inform you that your manuscript has been deemed suitable for publication in PLOS ONE. Congratulations! Your manuscript is now with our production department. 

Kind regards, 

on behalf of

Dr. Ji Yi 

Academic Editor

PLOS ONE